# Pathological Mechanisms in Sjögren’s Disease Likely Involve the ADP-Ribosyl Cyclase Family Members: CD38 and CD157

**DOI:** 10.3390/ijms262311544

**Published:** 2025-11-28

**Authors:** Michaela Rosecka, Martina Kolackova, Moeina Afshari, Eva Jozifkova, Radovan Slezak, Jan Krejsek, Vladimira Radochova

**Affiliations:** 1Department of Clinical Immunology and Allergology, Faculty of Medicine, Charles University, 500 03 Hradec Kralove, Czech Republicafsharim@lfhk.cuni.cz (M.A.);; 2Department of Biology, Jan Evangelista Purkyne University, 400 96 Usti nad Labem, Czech Republic; 3Department of Dentistry, Faculty of Medicine, Charles University, 500 03 Hradec Kralove, Czech Republicvladimira.radochova@lfhk.cuni.cz (V.R.); 4Department of Dentistry, University Hospital Hradec Kralove, 500 05 Hradec Kralove, Czech Republic

**Keywords:** Sjögren’s disease, CD38, CD157, ADP-ribosyl cyclase family

## Abstract

Peripheral blood serves both as a source of effector immune cells that migrate to exocrine glands and as a reflection of the immunological changes occurring in patients with Sjögren’s disease (SjD). These changes may be linked to the clinical state of these patients. We analyzed total cell counts in the peripheral blood, as well as frequencies of individual leukocyte subpopulations, membrane expression levels of CD38 and CD157, and serum concentrations of soluble sCD38 and sCD157 in SjD patients (*n* = 40) and age-matched healthy controls (*n* = 20). Hierarchical clustering based on the cell count of leukocyte subpopulations was employed to identify distinct patient subgroups. Associations between these clusters and clinical parameters were subsequently evaluated. Key findings included a reduction in lymphocyte counts and their subpopulations, alongside increased CD38 expression on CD38^+^ B cells (*p* = 0.047) and, unexpectedly, on monocytes (*p* = 0.014) when comparing patients and controls. The involvement of innate immunity was further supported by the differential expression of CD157 across patient samples. Patients with low cell counts exhibited reduced CD157 expression on monocytes and granulocytes (*p* < 0.02), tested positive for anti-Ro antibodies, and reported severe fatigue. Our findings suggest that innate immune cells, such as monocytes and granulocytes in peripheral blood, are also likely to contribute to the manifestation and progression of SjD. The differential expression of CD157 may reflect distinct immunopathological states and warrants further investigation, as its precise role in exocrine gland involvement and extra-glandular manifestations lies beyond the scope of this study.

## 1. Introduction

Sjögren’s disease (SjD) is a persistent autoimmune condition marked by immune cell infiltration and dysfunction of exocrine glands, particularly those producing saliva and tears. While hallmark symptoms include oral and ocular dryness, the disorder may extend to other glands or present with systemic features [1]. Despite a long-term interest in the underlying mechanisms, the roles of specific regulatory molecules remain incompletely understood.

Among these, CD38 and its paralog CD157 can be considered essential players in cell interactions. Both molecules engage in calcium-mediated intracellular communication, a fundamental process in immune activation [2,3,4]. CD38 and CD157 catalyze the conversion of NAD^+^ into adenosine diphosphate ribose (ADPR), cyclic ADPR (cADPR), and nicotinic acid adenine dinucleotide phosphate (NAADPR)—potent secondary messengers that regulate calcium efflux from intracellular stores (reviewed by Malavasi F. et al., 2008 [5]). Through their enzymatic activity, these molecules influence NAD^+^ metabolism, which is closely linked to inflammation and aging [6,7,8]. Primarily, CD38 expression increases with age and during inflammatory responses, suggesting a potential compounding effect in older individuals with autoimmune conditions, such as SjD [6,7,8,9,10]. Unsurprisingly, several inhibitors of CD38 have been shown not only to inhibit the secretion of inflammatory cytokines, such as IL-1β, but also to prevent age-related changes [11,12,13]. However, their efficacy in autoimmune suppression remains to be established [14].

Structurally, CD38 is a transmembrane protein, while CD157 is anchored to the plasma membrane via a glycosylphosphatidylinositol (GPI) linkage [15,16,17]. Despite these differences, both molecules participate in cell adhesion and migration—functions that are especially relevant in SjD, where immune cells infiltrate exocrine glands [18,19,20,21]. CD157 interacts with fibronectin and other extracellular matrix components [18], while CD38 binds to CD31 (Platelet Endothelial Cell Adhesion Molecule, PECAM-1) [22]. Their association with lipid rafts facilitates efficient signaling through pathways such as MAPK/ERK1/2 and PI3K/Akt, independent of calcium mobilization [21,23,24,25]. Additionally, both proteins can be shed into the extracellular space, where their soluble forms retain enzymatic and receptor functions and have been proposed as biomarkers in specific clinical conditions, including rheumatoid arthritis [26,27,28,29,30].

In Sjögren’s disease, alterations in the activation of immune cells contribute to abnormal immune responses, leading to characteristic symptoms such as reduced salivation and lacrimation [31]. Minor salivary gland (MSG) biopsy remains a key diagnostic tool, with lymphocytic infiltration, described as focus score, serving as an indicator of disease activity [1,32]. CD4^+^ T cells predominate in these infiltrates, and their elevated CD38 expression reflects an activated state [33]. B cells, plasma cells, monocytes, macrophages, and dendritic cells also contribute to the inflammatory milieu [33,34]. CD38 facilitates the migration of dendritic cells to sites of inflammation and is a defining feature of plasma cells, which localize in the interstitium and at the periphery of lesions [19,35,36,37]. Long-lived plasma cells, in particular, produce autoantibodies such as anti-muscarinic type 3 receptor, anti-Ro, and anti-centromere antibodies, which are implicated in glandular dysfunction and epithelial cell apoptosis [37,38,39,40].

The frequency and elevated counts of specific cell populations, the ratios of lymphocyte subpopulations, and the expression of markers, such as CXCL13, may be associated with lesion severity [34,41,42,43]. Notably, well-formed germinal center-like structures are considered highly significant due to their association with an increased risk of lymphoma [42].

While the lacrimal and salivary glands are most commonly studied, other exocrine glands—including vaginal, sweat, meibomian, pancreatic, and respiratory tract glands—are also affected [44,45,46,47,48]. These sites are often overlooked due to the accessibility and diagnostic utility of MSG biopsies. Consequently, the relationship between MSG involvement and broader exocrine or extra-glandular manifestations remains poorly defined. Symptoms such as vasculitis, neuropathy, and arthritis suggest systemic immune dysregulation (reviewed by Mihai A. et al., 2023, [49]). Underlying immune mechanisms likely include molecules such as CD38, which is widely expressed across various immune cell types. Consistently, the off-label use of daratumumab, a monoclonal antibody targeting CD38, has shown promising results in refractory cases of SjD [50].

In contrast, the role of CD157 in autoimmune diseases is less well characterized. It is known to regulate monocyte and neutrophil trafficking, B cell differentiation, and NAD^+^ metabolism [17,20,21,51,52]. Its expression on stem cells may also contribute to neural tissue regeneration [3,53]. CD157 has been implicated in several pathological conditions, primarily of neurological origin and cancer [30,54,55]. However, only a few studies have explored its role in autoimmune diseases [27], and its significance in the pathology of SjD remains largely unknown.

Given the systemic nature of Sjögren’s disease, we chose to analyze peripheral blood, as changes in this compartment may reflect broader immunological involvement beyond the most commonly affected exocrine glands. Peripheral blood sampling is a minimally invasive procedure that imposes little burden on patients, making it a practical choice for clinical research. Using a cost-effective and straightforward staining protocol with only four fluorochromes, we distinguished major immune cell populations—T cells, B cells, monocytes, and granulocytes—by flow cytometry. We compared cell counts, immune cell subset frequencies, expression levels of CD38 and CD157, and concentrations of their soluble forms (sCD38 and sCD157) between age-matched healthy controls and patients diagnosed with primary SjD. Furthermore, we assessed correlations between these immunological parameters and additional markers, including clinical parameters, within the patient cohort.

## 2. Results

### 2.1. Patients

Patients suffered from various additional symptoms that could be attributed to their primary autoimmune disorder. Besides local symptomatic treatment of dry eyes and mouth, most patients were on systemic pharmacological treatment (Table 1).

Eighty-five percent of patients had eyes afflicted by the disease, as confirmed by Schirmer’s test. Salivary glands were impaired (characterized by low salivation) in 60% of patients, and anti-Ro antibodies were detected in 87.5% of patients. Histological examination was performed in 62.5% of patients, of whom 96% were positive (that is, FS was ≥1) and 4% were negative (FS ≤ 1). The remaining patients did not undergo biopsy due to its invasive nature, which resulted in incomplete histological data for this cohort.

### 2.2. Leukocytes and Their (Sub)Populations

Although no statistically significant differences were found in the percentage of leukocyte populations (T cells, B cells, lymphocytes, monocytes, and granulocytes) (Appendix A), differences were observed in the absolute count of several leukocyte populations between patients and controls (Figure 1d–f). Patients displayed a lower count of T cells (1.25 × 10^9^/L vs. 1.86 × 10^9^/L, *p* = 0.019), B cells (0.16 × 10^9^/L vs. 0.25 × 10^9^/L, *p* = 0.012), and lymphocytes (1.76 × 10^9^/L vs. 2.45 × 10^9^/L, *p* = 0.007). Although the absolute counts of monocytes, granulocytes, and leukocytes appeared to be lower in patients than in controls, no statistically significant differences were observed (Figure 1a–c).

### 2.3. The Expression of CD157 and CD38 on the Cell Surface

The entire populations of CD14^+^ monocytes and granulocytes expressed CD157 (Appendix A), and its expression was quantified as median fluorescence intensity (MFI) (Figure 2a,b). Although CD157 appeared higher in patients compared to controls, the difference did not reach statistical significance.

Similarly, while the entire CD14^+^ monocyte population expressed CD38, only certain subsets of T cells and B cells were positive for this molecule (Appendix A). These CD38^+^ subsets were not further characterized; however, MFI was assessed and statistically compared exclusively within these CD38^+^ T and B cell populations (Figure 2c–e). Significant differences in CD38 expression between patients and controls were found in the B cell population (*p* = 0.047, MFI: 120 vs. 105, respectively, Figure 2d) and the monocyte population (*p* = 0.014, MFI: 364 vs. 312, respectively, Figure 2e).

In our study, many patients were receiving systemic pharmacological therapy, which allowed us to stratify them into subgroups based on treatment type (NSAIDs, corticosteroids, antimalarials, cyclosporin, and various combinations).

To address this, we performed statistical analyses comparing cell counts and CD38/CD157 expression between treated and untreated patients. These analyses revealed no significant differences, except for the effect of corticosteroids on CD38 expression in CD38^+^ B cells. Specifically, CD38 expression was significantly higher in patients not receiving corticosteroids compared to both healthy controls and corticosteroid-treated patients (*p* < 0.01 for both comparisons). Importantly, patients treated with corticosteroids did not differ from healthy controls, as illustrated in Figure 3.

### 2.4. Serum Level of sCD157 and sCD38

The concentration of sCD157 was in the nanogram values in both patients and controls (Figure 4a). Soluble CD38 exhibited varying values, ranging from undetectable concentrations (in most samples) to thousands of picograms (in a few samples) in both groups (Figure 4b). No significant differences were found for either sCD157 or sCD38.

### 2.5. Hierarchical Clustering

Patients were grouped by hierarchical clustering using Ward’s method based on their T cell, B cell, monocyte, and granulocyte counts. At first patients were grouped into four distinct groups, Figure 5; however, they were re-grouped due to the low frequencies in the groups just into two groups; the first ‘low-count group’ consisting of 24 patients who had low count of any subpopulation of leukocytes, and the second ‘higher-count’ group containing 16 patients whose at least one subpopulation was of higher count than the first group as displayed in Figure 6.

Expressions of CD157 and CD38, as well as several clinical parameters, including medical treatments, were compared between these two groups.

Both groups differed significantly only in the expression of CD157 in monocytes and granulocytes, while other differences in expression were not statistically significant (Figure 7). The higher-count group expressed CD157 with higher intensity than the lower-count group (Figure 7a,b).

Differences were observed in the frequency of individuals experiencing fatigue and the presence of autoantibodies (anti-Ro), with *p*-values of 0.015 and 0.007, respectively, when comparing the two groups. All patients with low counts in leukocyte subpopulations had autoantibodies present and, in addition, experienced fatigue more frequently than patients in the other group (Table 2 and Table 3).

## 3. Discussion

Sjögren’s disease (SjD) belongs to the group of autoimmune connective tissue disorders, alongside systemic lupus erythematosus, rheumatoid arthritis, scleroderma, and polymyositis. Although various hypotheses exist regarding their initiation, the exact triggers remain elusive. A hallmark of these conditions is the breakdown in immune regulation.

Historically, mononuclear infiltrates have been the primary focus in immunophenotyping studies of affected salivary glands in SjD. Meanwhile, other findings have underscored the active role of salivary gland epithelial cells (SGEC) in modulating local immune responses, particularly through interactions with lymphocytes, as reviewed by Tang Y. et al., 2024 [56]. SGECs facilitate B-cell recruitment, activation, and survival [57,58], and also seem to induce monocyte recruitment and differentiation into dendritic cells [59]. Immune cells, originating from peripheral blood, can migrate to inflamed tissues upon activation [60,61]. Direct observations of peripheral blood cells migrating to inflamed lesions in humans, particularly in patients with Sjögren’s disease, are lacking. However, this assumption is supported by studies that report correlations between specific cell populations in peripheral blood and lymphocytic foci [62,63]. As a marker of cell activation, the systemic production of cytokines, such as IFN-γ, IL-6, and IL-10, has been observed in patients with the disease [64,65]. The therapeutic efficacy of IL-6 blockade using tocilizumab in certain cases further highlights the relevance of these cytokines [66,67]. Despite their diagnostic potential (some of them have even been linked to the production of autoantibodies), cytokines are not routinely measured in clinical settings. Yet, some studies have reported soluble BAFF levels in patient cohorts [68].

In our study, we explored whether soluble CD38 and CD157 levels correlate with disease status. Unfortunately, no significant differences were observed between patients and healthy controls. We also examined peripheral blood cell counts and frequencies, along with the altered expression of membrane-bound regulatory molecules, hypothesizing that these parameters might reflect systemic immune activation. Consistent with the findings by Davies et al. (2017), we observed a reduction in total lymphocyte counts in patients [69], including both T and B cells (Figure 1). Interestingly, monocyte and granulocyte counts were also slightly reduced, diverging from Davies’s report of increased CD16^+^ granulocytes in patients with extra-glandular involvement. We did not detect significant changes in cell frequencies, possibly due to our broad phenotyping approach, which focused on major leukocyte populations [70]. Nonetheless, other studies have reported altered frequencies of classical and non-classical monocytes (CD14^low^ CD16^+^) in SjD [59,71].

Aberrant expression of surface markers has been documented in SjD. For example, transcriptomic profiling by He Y. et al. (2022) identified TRAIL expression on monocytes as a distinguishing feature [72], while Yoshimoto K. et al. (2020) implicated elevated BAFF receptor expression in promoting IgG overproduction (2020) in patients with SjD [73]. These findings suggest that monocytes, alongside lymphocytes, contribute to disease pathogenesis. CD38 has emerged as a key regulatory molecule in this context; however, our study suggests that CD157 may also be involved in the pathogenesis of Sjögren’s disease.

The ADP-ribosyl cyclase family is involved in cell metabolism, executive cell functions, and cell interactions by generating the calcium-mobilizing second messengers, cADPR and NAADPR [74,75]. This mechanism is so fundamental that ADP-ribosyl cyclases can be found in even the most basic unicellular organisms, plants, and animals, which are evolutionarily distant from humans [76]. In mammals, CD38 and CD157 are the two characterized members of this enzyme family. Unlike most ADP-ribosyl cyclases, which are typically intracellular, CD38 and CD157 are ectoenzymes anchored to the plasma membrane—a feature that introduces a topological paradox, as their catalytic domains face the extracellular space, as reviewed by Malavasi F. et al. (2008) and De Flora A. et al. (1997) [5,77]. Intracellular localization of CD38 has been reported and may partially resolve this paradox [78]. CD38 and CD157 display not only cyclase activity but mainly NAD^+^ glycohydrolase enzymatic activity [52,79]. Several enzymatic products have been described, but the chief metabolites comprise ADPR, cADPR, and NAADPR, all of which serve as calcium-releasing secondary messengers [52,74,79,80].

Beyond their enzymatic roles, CD38 and CD157 also act as receptors [18,22]. Their membrane localization enables them to mediate cell adhesion and migration, and receptor cross-linking initiates downstream signaling pathways, adding another layer of immune regulation [19,21,81,82].

Due to its broad regulatory roles and extensive expression profile, CD38 has become a focal point in studies of various human diseases [83]. It is widely expressed across hematopoietic cells, encompassing both lymphoid and myeloid lineages. Within T and B lymphocytes, CD38 expression levels fluctuate depending on the cell’s developmental stage and activation status [84]. The molecule is also present in natural killer (NK) cells, monocytes, macrophages, granulocytes, and dendritic cells [85]. Notably, CD38 expression is not confined to immune cells; it has also been identified in non-hematopoietic tissues, including epithelial and neural cells [86,87]. CD157, although more restricted in its distribution, is predominantly found in myeloid immune cells such as monocytes and neutrophils, and has also been detected in neural tissue [54,88].

Several investigations have examined CD38 expression in the context of SjD, focusing on either the proportion of CD38-positive cells or the intensity of expression, sometimes stratifying patients based on autoantibody profiles, such as anti-Ro positivity [69,71,89]. In our study, we assessed CD38 and CD157 expression in lymphocytes and monocytes from a geographically and ethnically homogeneous cohort of unstratified SjD patients and healthy controls. We observed a significant upregulation of CD38 in CD38^+^ B cells and monocytes (Figure 2d,e), suggesting increased activation of these populations. CD157 expression was also elevated in monocytes and granulocytes, although the difference did not reach statistical significance, likely due to its bimodal distribution among patients (Figure 2a,b). Clustering analysis corroborated the differences in CD157 expression across the patients’ samples (Figure 7). Individuals with low CD157 expression in monocytes and granulocytes formed a distinct cluster characterized by reduced leukocyte counts, pronounced fatigue, and the presence of anti-Ro antibodies. This subgroup also tended to show lower CD38 expression in monocytes. These findings may indicate more profound alterations in innate immune function, potentially reflecting monocyte exhaustion or dysregulation in SjD. However, this hypothesis requires further validation through targeted ex vivo functional studies.

While the real-time analysis of CD38 and CD157 enzymatic or signaling activity remains technically challenging, several studies have focused on the expression of CD38 in immune cells in SjD, and our study complements them with the observation of CD157 expression. Although it is a descriptive research study in nature, it still offers insight into the aberrant regulatory mechanisms underlying Sjögren’s disease.

## 4. Materials and Methods

### 4.1. Participants

All patients met the American-European Consensus Criteria (AECC) for the diagnosis of Sjögren’s disease [90]. The cohort consisted of patients with primary Sjögren’s syndrome, as defined by oral and ocular signs and symptoms, as well as laboratory results, in accordance with AECC recommendations. Patients who did not agree to a lip biopsy had either a sign of oral and ocular dryness and positive anti-Ro antibodies or were excluded from the analysis. Examination and diagnosis were performed at the University Hospital in Hradec Kralove, Czech Republic.

Patients with Sjögren’s syndrome (‘Patients’, *n* = 40) and a group of healthy individuals (‘Controls’, *n* = 20) enrolled in this research project were of Caucasian origin. They came from the same geographical area within the Czech Republic. Their demographic and clinical data are summarized in Table 4.

### 4.2. Flow Cytometry

The analysis was performed on whole-blood samples, which were collected into Vacutainer tubes treated with sodium heparin (Becton Dickinson, Franklin Lakes, NJ, USA). The total number of leukocytes (leukocyte count) was calculated using a hemocytometer. We stained blood samples using monoclonal, fluorochrome-conjugated antibodies to distinguish populations and subpopulations of leukocytes as well as to quantify cell-surface expression of CD38 and CD157. All the antibodies were purchased from a commercial vendor, Exbio (Czech Republic): anti-human CD3 APC (clone MEM-57), anti-human CD19 PerCP (clone LT19), anti-human CD14 FITC (clone MEM-18), anti-human CD38 PE (clone HIT2), and anti-human CD157 PE (clone SY11B5). CD38 and CD157 were stained in separate tubes to prevent cross-staining due to the high level of sequence homology between the two proteins. The antibodies were added to 50 μL of the blood sample (adjusted to 5 × 10^9^ cells/L) at the concentration recommended by the manufacturer. Following staining, red blood cells were lysed under isotonic conditions using OptiLyse C (Beckman Coulter, Indianapolis, IN, USA). Excess of antibodies as well as cell debris were washed off by centrifugation. The acquisition was performed using the Lyse/Wash protocol in CellQuest Pro 4.02 software on a FACSCalibur flow cytometer (Becton Dickinson, Franklin Lakes, NJ, USA) immediately after blood sample collection and staining. The batch analysis was performed retrospectively using FlowJo™ 10 software (Becton Dickinson, Franklin Lakes, NJ, USA). The gating strategy is depicted in the Appendix A. The intensity of expression of CD38 and CD157 was evaluated only in cells expressing these molecules. Although all monocytes and granulocytes expressed CD38 and CD157, only certain subsets of T cells and B cells were positive for CD38. Therefore, all descriptions and analyses in this study refer specifically to CD38^+^ T and B cell populations.

### 4.3. Enzyme-Linked Immunosorbent Assay (ELISA)

A pair of antibodies, a recombinant protein standard, and streptavidin-HRP for the detection of either sCD38 or sCD157 were bought from Bio-Techne (Minneapolis, MN, USA). Overnight coating of the plates, as well as subsequent steps, was performed according to the manufacturer’s manual. The absorbance was measured on BioTek Synergy HTX Multimode Reader (Agilent Technologies, Santa Clara, CA, USA) at 450 nm, and the resulting concentrations were calculated using Gen5 software. The minimal detection values of sCD38 and sCD157 in this assay were 31.2 pg/mL and 39.1 pg/mL, respectively. sCD38 and sCD157 were detected in serum samples that had been stored at −80 °C before analysis.

### 4.4. Statistical Evaluation

The comparison of the patient and control groups was performed using the following tests in Statistica 14 (TIBCO Statistica, Palo Alto, CA, USA) and Excel (Microsoft Office 365, Microsoft Corporation, Redmond, WA, USA). Normality was assessed using the Shapiro–Wilk test, and equality of variances was evaluated with the F-test. Numerical data were analyzed using the Brunner-Munzel test to compare the patient and control groups. In a few cases, other tests were used, as mentioned in the Figures. Fisher’s exact test was used for categorical data. Differences were considered statistically significant when *p* ≤ 0.05. The graphical display was created using GraphPad Prism 8 software (GraphPad Software, Boston, MA, USA). The results of statistical analyses are included in the description of the individual figures. The statistics display medians, marked by larger lines, and interquartile ranges, indicated by smaller lines.

Hierarchical clustering using Ward’s method was performed to divide the patient group according to the Z-score-standardized cell numbers in individual subpopulations of leukocytes. SPSS Statistics 29, IBM (Armonk, NY, USA), was used for clustering, cluster plotting, and subsequent analyses. Analyses are included in tables and figures. A 5% significance level has been used to reject the null hypotheses.

## 5. Conclusions

In this study, we present the first comparative analysis of CD38 and CD157 expression in the peripheral blood of SjD patients. Our findings reveal that while CD38 and CD157 are generally upregulated in the monocytes and granulocytes of affected individuals, their expression is paradoxically reduced in those with more severe clinical manifestations, underscoring a complex regulatory role and involvement of innate immunity that warrants further investigation. The expression pattern of these molecules also suggests potential links to immune cell activation and NAD^+^ metabolism, further reinforcing their relevance in disease pathophysiology.

## Figures and Tables

**Figure 1 ijms-26-11544-f001:**
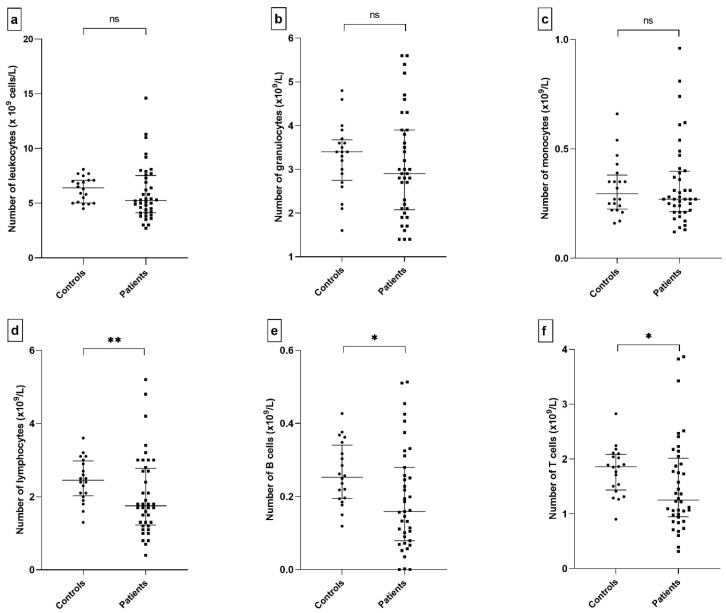
Leukocyte count and leukocyte (sub)populations in patients and controls. The total count of cell populations was calculated as a ratio of a given population (measured by flow cytometry) in the leukocyte count. Comparison of leukocytes (*p* = 0.095) (**a**), granulocytes (*p* = 0.416) (**b**), monocytes (*p* = 0.849) (**c**), lymphocytes (*p* = 0.007) (**d**), B cells (*p* = 0.012) (**e**), and comparison of T cells (*p* = 0.019) (**f**). Statistical significance was indicated as follows: *p* ≤ 0.05 (*), *p* ≤ 0.01 (**), and non-significant differences (*p* > 0.05) as ns.

**Figure 2 ijms-26-11544-f002:**
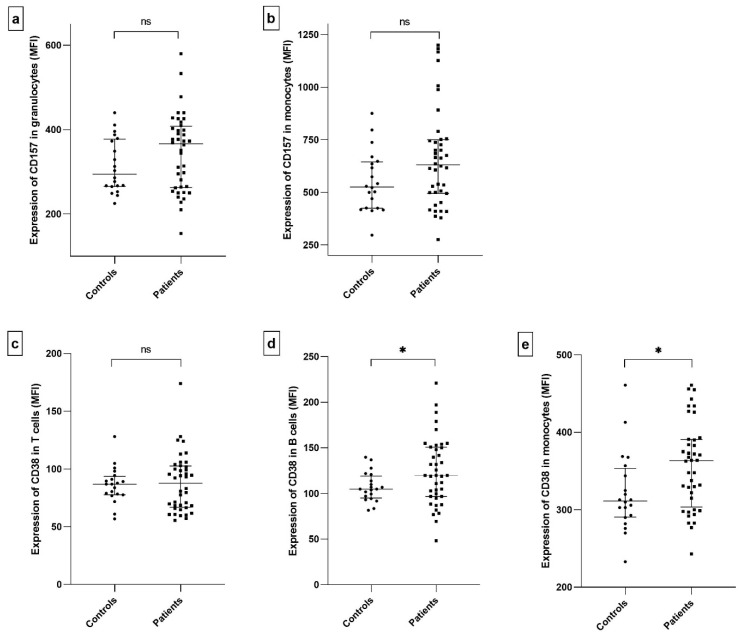
Expression of CD157 and CD38 in CD157^+^ and CD38^+^ cell populations from patients and controls. Expression levels were quantified as median fluorescence intensity (MFI) in cells positive for the respective molecule. Comparisons include: CD157 in granulocytes (*p* = 0.215) (**a**), CD157 in monocytes (*p* = 0.107) (**b**), CD38 in T cells (*p* = 0.948) (**c**), CD38 in B cells (*p* = 0.047) (**d**), and CD38 in monocytes (*p* = 0.014) (**e**). Statistical significance was indicated as described in Figure 1 (*p* ≤ 0.05 (*), and non-significant differences (*p* > 0.05) as ns).

**Figure 3 ijms-26-11544-f003:**
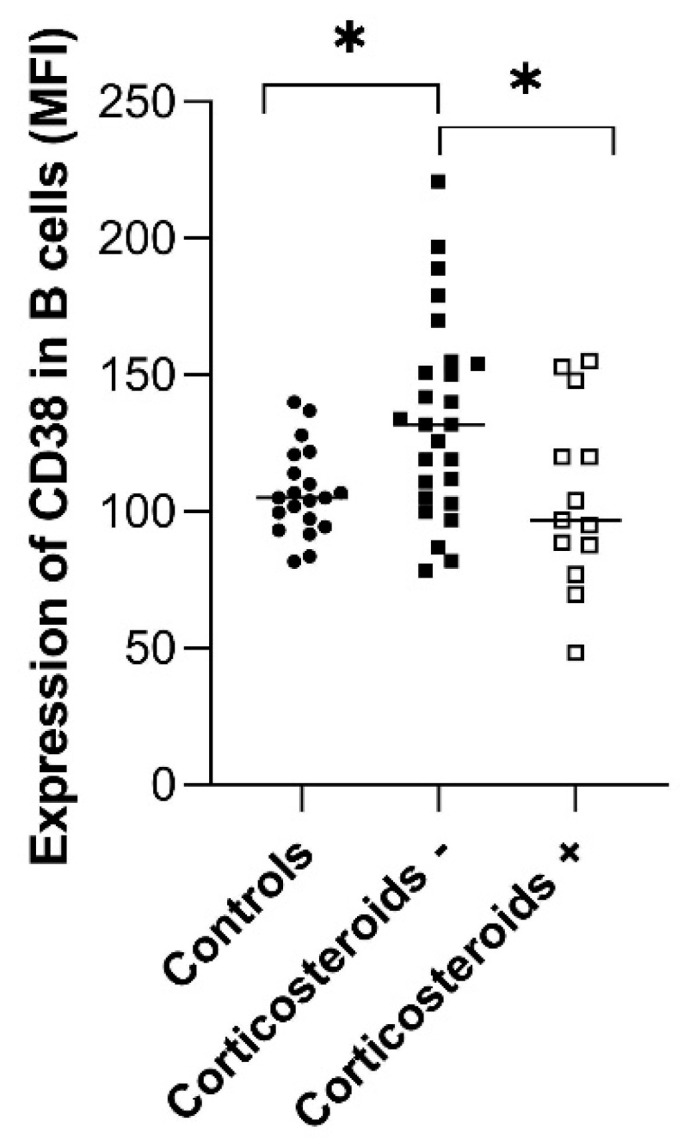
Effect of corticosteroid treatment on CD38 expression in CD38^+^ B cells. Patients not receiving corticosteroids (*n* = 27) showed significantly higher expression compared to both patients on corticosteroid therapy (*n* = 13, *p* = 0.03) and healthy controls (*n* = 20, *p* = 0.027), as determined by the Kruskal–Wallis test with multiple comparisons; *p* ≤ 0.05 (*).

**Figure 4 ijms-26-11544-f004:**
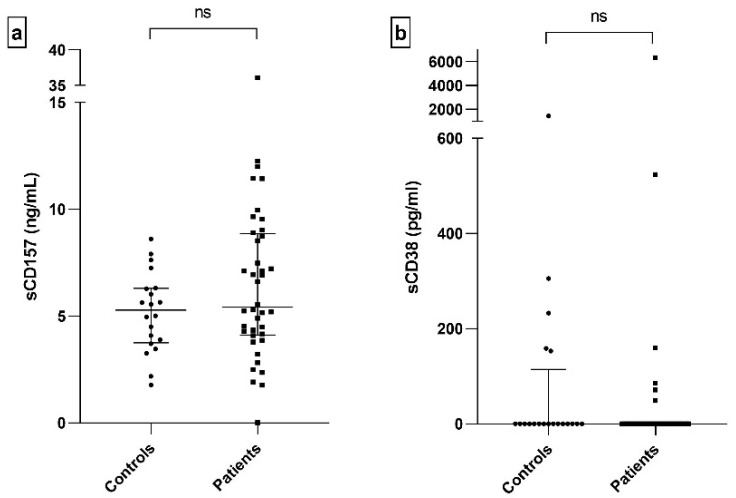
Concentration of sCD157 and sCD38 in serum of patients and controls. Comparison of sCD157 (*p* = 0.229) (**a**) and sCD38 (*p* = 0.355) (**b**); ns: non-significant.

**Figure 5 ijms-26-11544-f005:**
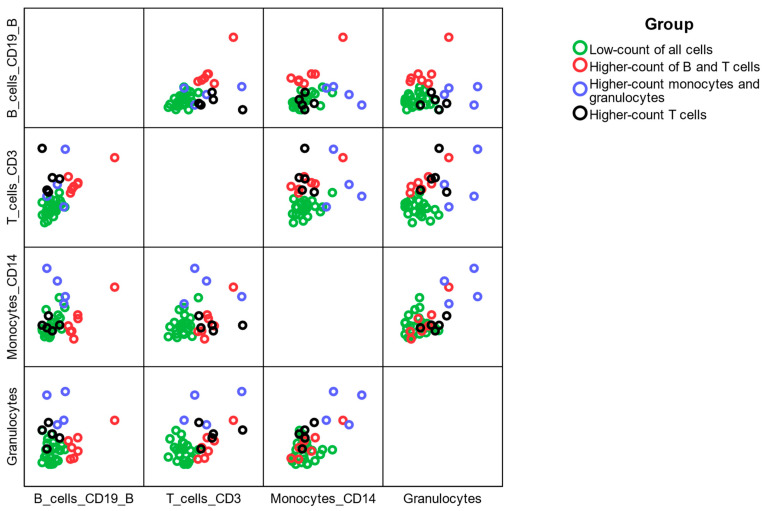
Four clusters of patients were identified according to their cell counts in the subpopulations of leukocytes. Low-count group (*n* = 24), higher-count of B and T cells group (*n* = 7), higher-count of monocytes and granulocytes (*n* = 4), and higher-count of T cells (*n* = 5).

**Figure 6 ijms-26-11544-f006:**
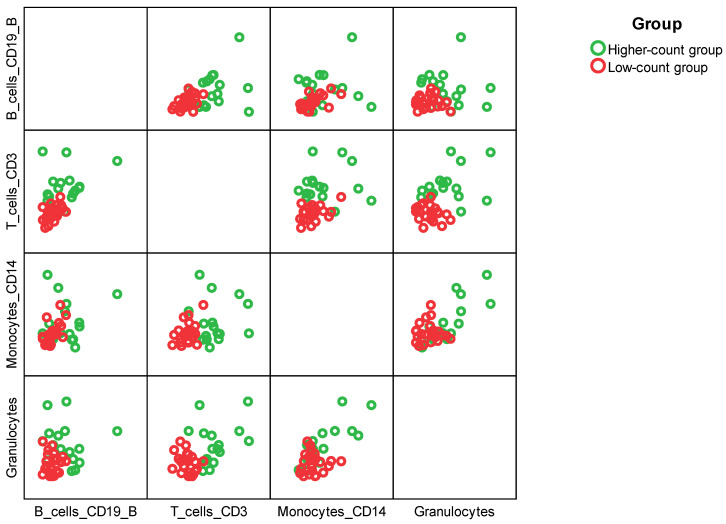
Two clusters of patients were identified based on their cell counts in the subpopulations of leukocytes: a low-count group (*n* = 24) and a high-count group (*n* = 16).

**Figure 7 ijms-26-11544-f007:**
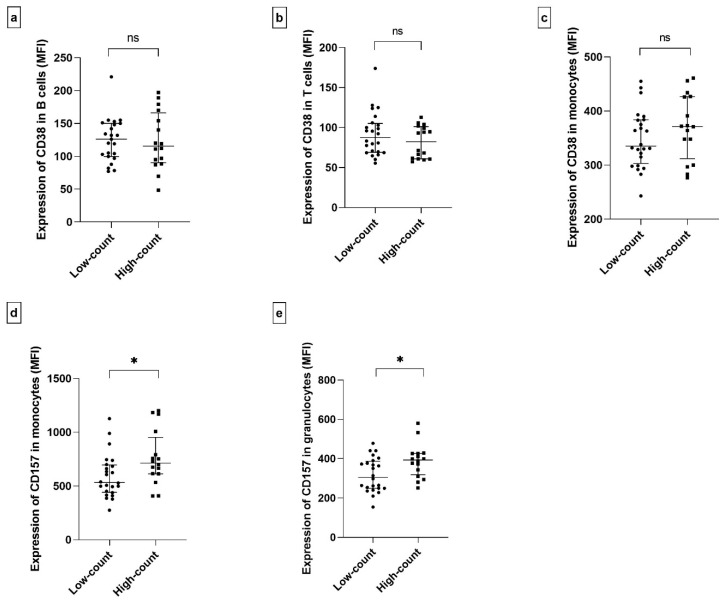
Comparison of the expressions of CD157 and CD38 between the low-count group (*n* = 24) and the higher-count group (*n* = 16). Comparison of CD38 in CD38^+^ B cells (*p* = 0.845) (**a**), CD38 in CD38^+^ T cells (*p* = 0.272) (**b**), CD38 in monocytes (*p* = 0.284) (**c**), CD157 in monocytes (*p* = 0.017) (**d**), and CD157 in granulocytes (*p* = 0.012) (**e**). Statistical significance was indicated as described in Figure 1 (*p* ≤ 0.05 (*), and non-significant differences (*p* > 0.05) as ns).

**Table 1 ijms-26-11544-t001:** Extra-glandular manifestations and treatment.

Patients (40)	No. (Frequency)
**Additional symptoms**	
TMJ disorder	20 (50%)
Thyroid dysfunction	11 (27.5%)
Arthritis	25 (62.5%)
Weight loss	9 (22.5%)
Fatigue	27 (67.5%)
**Systemic pharmacological treatment**	
NSAIDs	11 (27.5%)
Cyclosporin A	5 (12.5%)
Corticosteroids	13 (32.5%)
Antimalarials	11 (27.5%)
Combined treatment	8 (20%)

Combined treatment involved either a combination of Cyclosporin A and corticosteroids or the combination of antimalarials and corticosteroids. Abbreviations: TMJ disorder: Temporomandibular joint disorder, NSAIDs: Non-steroidal anti-inflammatory drugs.

**Table 2 ijms-26-11544-t002:** Autoantibodies.

	Yes	No
Low-count group (24)	24 (100%)	0 (0%)
Higher-count group (16)	11 (68.8%)	5 (31.3%)

Number of patients in each category, with a 2-sided Chi-Square comparison, *p* = 0.007. Relative risk (RR) = 1.46; 95% Confidence interval (CI): 1.05–2.02; *p* = 0.026.

**Table 3 ijms-26-11544-t003:** Fatigue.

	Yes	No
Low-count group (24)	20 (83.3%)	4 (16.7%)
Higher-count group (16)	7 (43.8%)	9 (56.3%)

Number of patients in each category, Chi-Square 2-sided comparison, *p* = 0.015. RR = 1.9; 95% CI: 1.06–3.42; *p* = 0.031; description as in Table 2.

**Table 4 ijms-26-11544-t004:** Demographic and clinical data.

	Controls (20)	Patients (40)
Men/Women (no.)	1/19	3/37
Age (years)	54	55.5
No. of leukocytes (×10^9^/L)	6.4	5.25
**Oral symptoms and signs, and their laboratory evaluation**		
Xerostomia (no.)	0	33
Dysphagia (no.)	0	23
Decreased salivary flow (no.)	0	24
MSG biopsy (no.)	N/A	24 (15)
**Ocular symptoms and signs**		
Xerophtalmia (no.)	0	34
Schirmer’s test (no.)	N/A	34
**Presence of anti-Ro autoantibodies**	0	35

Age and number of leukocytes are displayed as median values. The number of positive cases describes all other parameters. Statistical evaluation of demographic data and leukocytes is in the Appendix A and Figure 1a. Definitions: A Positive test of decreased salivary flow was defined as ≤1.5 mL of unstimulated saliva in 15 min. A positive Schirmer’s test was evaluated as tear production of ≤5 mm on the testing paper in 5 min. MSG (Minor Salivary Gland) biopsy was positive when the focus score (FS) was ≥1 (while FS was defined as a count of lymphocytic foci containing more than 50 mononuclear cells per 4 mm^2^ tissue biopsy). Fifteen patients did not sign the agreement for the biopsy. The presence of anti-Ro and anti-La (autoantibodies against ribonucleoprotein) was detected by immunoblotting and enzyme-linked immunosorbent assay. Xerophthalmia and xerostomia were characterized as the subjective perception of eye and mouth dryness, respectively.

## Data Availability

The original contributions presented in this study are included in the article.

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
