# Peer review of "Pathological Mechanisms in Sjögren’s Disease Likely Involve the ADP-Ribosyl Cyclase Family Members: CD38 and CD157"

_ijms, 2025, doi:10.3390/ijms262311544_

Round 1

Reviewer 1 Report

Comments and Suggestions for Authors

The article titled ‘Pathological Mechanisms in Sjögren’s Disease Likely Involve the ADP-Ribosyl Cyclase Family Members: CD38 and CD157’ focuses on revealing the immunological mechanism that’s associated with SjD by analyzing the peripheral blood.

This manuscript is of great focus, and I especially appreciate the hierarchical clustering of different groups, which provides vital information regarding the disease immunological profiles.

Although I found that the manuscript is enriched with a thorough introduction and a valid rationale, I could see that there are some major adjustments needed.

  1. In the description of the patients selected in the study, the authors mentioned that most of the patients are undergoing systemic pharmacological treatment. Statistical analysis addressing the systemic treatment's effect on the results is needed while the treatment alters immune profiling (PMID: 9858064, PMID: 15716327, PMID: 26972611, PMID: 22734582).
  2. While this study involves flow cytometry heavily, it is recommended to better justify the gating.
    1. Singlet gating is required.
    2. In current gating, granulocytes and monocytes are potentially polluted with neutrophils and doublets.
    3. Regarding the expression of CD157 and CD38 on the cell surface, according to the Supplementary Figure B, the MFI of CD38 for T cells and B cells is only applied to the CD38+ population, and does not reflect the MFI of the whole population of T cells or B cells.
  3. Since the study focuses on the peripheral blood samples, a discussion of how relevant the results are to the local disease loci is needed.
  4. One minor comment: in the section ‘2.2. Leukocytes and their (sub)populations’, in the first sentence, the authors cited Supplementary Figure A, which is not related to the content.

Author Response

Dear Reviewer,

Thank you for your thoughtful comments and suggestions. Please find our detailed responses in the attachment

Reviewer 2 Report

Comments and Suggestions for Authors

The study aims to investigate the role of CD38 and CD 157 in the clinical state of Sjorgen's disease. The authors revealed significant low CD157 expression in low counted cells patients compared to high counted cells patients, however the CD38 tends to be higher. 

The manuscript is clear, well structured and with updated citations. The results are reproeducible based on the details given in methods. The figures and tables show data properly and easy to interpret. The conclusion is consistent with evidence and arguments presented.

Major comments:

-I suggest listing the scoring system of this diseases in the introduction. 

-Since one of the study goal is to show the correlation between expression of CD157 and clinical state of patient , so I think that authors should run the correlation test between  anti RO levels CD157 expression, or Chqi-square test  between  clinical observation other than fatigue (acc to the scoring system) and  the CD157 expression. 

Author Response

Dear Reviewer,

We are grateful for your encouraging remarks and for highlighting important aspects that strengthened our manuscript. Thank you for recognizing the value of our work and for providing thoughtful feedback. Please find our detailed response below:

Histological examination of lymphocytic foci is indeed an essential component of the scoring system. Unfortunately, many patients do not consent to such an invasive procedure, which resulted in missing data—a recognized limitation of our study. To address this, we substituted histological assessment with other clinical indicators, including anti-Ro antibody titers, ocular and oral symptoms, and objective signs. Nevertheless, we acknowledge that this approach compromises the completeness of patient scoring. For this reason, the scoring parameter was excluded from hierarchical clustering and subsequent statistical analyses.

Beyond anti-Ro antibody titers and fatigue, we compared additional clinical features between patients classified into the “high-count” and “low-count” clusters. These included xerostomia, xerophthalmia, dysphagia, arthritis, thyroid disease, wasting, positive Schirmer’s test, decreased salivary flow, and medical treatments. Pearson chi-square tests (two-sided) were applied to all comparisons. Significant differences were observed for fatigue (p = 0.0015) and high anti-Ro antibody titers (p = 0.007), while other parameters did not reach statistical significance. While inclusion of histological scoring could have provided additional insight into associations between histological severity and cell-surface marker expression, the absence of this parameter does not affect the validity of our main findings, which are based on peripheral blood immune cell phenotyping.

Round 2

Reviewer 1 Report

Comments and Suggestions for Authors

I appreciate the authors’ effort in point-to-point responses and editing. Just one more comment regarding the CD38 MFI analysis. In the manuscript, the second paragraph under section ‘2.3. The expression of CD157 and CD38 on the cell surface, the authors claimed that ‘the expression of the cell surface CD38 was analyzed in T cells, B cells, and monocytes (Figure 2c, d, e)’. As I pointed out in the previous review, the MFI of CD38 shown in the figures, based on the supplementary gating figures, are not in T cells, B cells nor monocytes but CD38+ T cells, CD38+ B cells, and CD38+ monocytes. As the authors wrote in the response, if ‘Including CD38-negative cells would have artificially lowered population MFI and biased the statistics, ’ which I’m slightly concerned about, since analyzing only the positive populations could potentially bias the statistics, the authors should either correct the manuscript or change the CD38 MFI analysis to include the CD38-negative populations. The same comment applies to the CD157 MFI analysis, even though all/majority of the monocytes and granulocytes are CD157 positive. The consistency in the gating and the manuscript is needed.

Author Response

The expression of CD157 and CD38 on the cell surface, the authors claimed that ‘the expression of the cell surface CD38 was analyzed in T cells, B cells, and monocytes (Figure 2c, d, e)’. As I pointed out in the previous review, the MFI of CD38 shown in the figures, based on the supplementary gating figures, are not in T cells, B cells nor monocytes but CD38+ T cells, CD38+ B cells, and CD38+ monocytes.....

Dear Reviewer,

Thank you for your valuable comment. If I understand correctly, your concern was that the expression of CD38 and CD157 (evaluated as median fluorescence intensity, MFI) was assessed only within cell populations positive for the respective marker, and this was not explicitly stated in the manuscript. This clarification is particularly important for T- and B-cell populations, as only certain subsets express CD38.

To address this, I have revised the manuscript, including supplementary files, to clearly indicate that MFI was analyzed exclusively in CD38⁺ or CD157⁺ cells. These changes are highlighted in yellow in the revised manuscript.

Best regards,

Martina Kolackova

Reviewer 2 Report

Comments and Suggestions for Authors

The authors have approperiatly addressed the first comment, and incorporated the missing statistical analyses for second comment.  

Author Response

The authors have approperiatly addressed the first comment, and incorporated the missing statistical analyses for second comment.  

Dear Reviewer,

Thank you for reviewing our manuscript and for confirming that no further changes are needed. We appreciate your time and effort in evaluating our work.

Best regards,

Martina Kolackova
